Transcriptomic analysis and physiological characteristics of exogenous naphthylacetic acid application to regulate the healing process of oriental melon grafted onto squash

Xu Chuanqiang chuanqiang79@syau.edu.cn 1 2 3
Wu Fang 1 2 3
Guo Jieying 1 2 3
Hou Shuan 1 2 3
Wu Xiaofang 1 2 3
Xin Ying 1 2 3
1 National & Local Joint Engineering Research Center of Northern Horticultural Facilities Design & Application Technology (Liaoning) , Shenyang , China
2 Key Laboratory of Protected Horticulture (Shenyang Agricultural University) Ministry of Education , Shenyang , China
3 College of Horticulture, Shenyang Agricultural University , Shenyang , China
Azeem Farrukh
Electronic publication date: 2022 Sep 15
Publication date: 2022
Volume: 10
Electronic Location ID: e13980
Received 2022 Jun 7; Accepted 2022 Aug 10
Copyright: ©2022 Xu et al.
Copyright year: 2022
Copyright holder: Xu et al.
License: This is an open access article distributed under the terms of the Creative Commons Attribution License, which permits unrestricted use, distribution, reproduction and adaptation in any medium and for any purpose provided that it is properly attributed. For attribution, the original author(s), title, publication source (PeerJ) and either DOI or URL of the article must be cited.
License URL: https://creativecommons.org/licenses/by/4.0/

Keywords: Oriental melon, Squash, Graft, Exogenous naphthylacetic acid, Transcriptomic analysis, Endogenous hormone, Signal transduction, ROS scavenging, Vascular bundle formation

Funding: The National Key Research and Development Program of China 2020YFD1000300 The National Natural Science Foundation of China 31401917 The China Agriculture Research System of Watermelon and Melon CARS-25 The Basic Research Project of Liaoning Province LSNJC202005 This research was funded by the National Key Research and Development Program of China (2020YFD1000300), the National Natural Science Foundation of China (31401917), the China Agriculture Research System of Watermelon and Melon (CARS-25), and the Basic Research Project of Liaoning Province (LSNJC202005). The funders had no role in study design, data collection and analysis, decision to publish, or preparation of the manuscript.

==============================
The plant graft healing process is an intricate development influenced by numerous endogenous and environmental factors. This process involves the histological changes, physiological and biochemical reactions, signal transduction, and hormone exchanges in the grafting junction. Studies have shown that applying exogenous plant growth regulators can effectively promote the graft healing process and improve the quality of grafted plantlets. However, the physiological and molecular mechanism of graft healing formation remains unclear. In our present study, transcriptome changes in the melon and cucurbita genomes were analyzed between control and NAA treatment, and we provided the first view of complex networks to regulate graft healing under exogenous NAA application. The results showed that the exogenous NAA application could accelerate the graft healing process of oriental melon scion grafted onto squash rootstock through histological observation, increase the SOD, POD, PAL, and PPO activities during graft union development and enhance the contents of IAA, GA3, and ZR except for the IL stage. The DEGs were identified in the plant hormone signal-transduction, phenylpropanoid biosynthesis, and phenylalanine metabolism through transcriptome analysis of CK vs. NAA at the IL, CA, and VB stage by KEGG pathway enrichment analysis. Moreover, the exogenous NAA application significantly promoted the expression of genes involved in the hormone signal-transduction pathway, ROS scavenging system, and vascular bundle formation.

Introduction

Grafting is a widely used technology to help horticultural plants overcome their limiting growth and reproduction factors (Lee et al., 2010). In cultivation, the most frequently consumed horticulture plants, such as melon, watermelon, tomato, eggplant, and cucumber, are often grafted (Pina & Errea, 2005; Zheng et al., 2010). Successful grafting is a complex process involving the initial adhesion of scion and rootstock, callus tissue formation, and vascular bundle connection during the graft healing process. Callus tissue formation at the graft interface initiates the grafting process, and a lack of callus tissue formation can lead to grafting failure (Pina & Errea, 2005). In addition, the callus cells differentiating into vascular tissue to re-connect the xylem and phloem at the graft junction is an essential process for successful grafting (Flaishman et al., 2008). A functional vascular connection is an essential signal for the establishment of successful graft healing (Turquois & Malone, 1996; Fernández-garcía, Carvajal & Olmos, 2004). However, the reconnection of the rootstock and the scion has tissue asymmetry. The phloem junction occurs at the graft junction before the xylem (Melnyk et al., 2015).

As we all know, complex physiological metabolites occur during the graft healing process (Koepke & Dhingra, 2013). Even though studies on the histological analysis and physiological and biochemical changes of graft healing have been reported (Chen et al., 2017; Cookson et al., 2013), the comprehensive understanding of the molecular mechanism of successful graft healing remains insufficient. The genome-wide transcriptome analysis could help to clarify the specific and underlying molecular mechanisms of grafting-depended biochemical processes. Some studies showed that transcriptomics efficiently analyzed the graft healing process of the different species. In the graft healing process of grapevine, transcriptional changes were examined via whole-genome microarray analysis, and the report revealed many genes associated with cell wall modification, hormone signaling, secondary metabolism, and wound responses (Cookson et al., 2013). Transcriptomic analysis of graft healing in Litchi showed that nine annotated unigenes that participated in the auxin signaling pathway had higher expression levels in the compatible grafts than incompatible ones (Chen et al., 2017). In the graft healing process of hickory, the transcriptomic analysis revealed 112 candidate unigenes, which participated in the auxin and cytokinin signaling pathways (Qiu et al., 2016). Genome-wide transcriptome analysis of tissues above and below graft junctions revealed that the inter-tissue communication process occurred independently of functional vascular connections and acted as a signal to activate vascular regeneration (Melnyk et al., 2018).

Phytohormones play an essential role in plant wound healing and vascular formation (Melnyk et al., 2015). The exogenous auxin application to callus enhanced the formation of xylem and phloem (Wetmore & Rier, 1963; Scarpella et al., 2006). The application of exogenous phytohormones, such as heteroauxin and zeatin, positively accelerated the healing by increasing the vascular bundle formation rate (Lu & Song, 1999). In the grafted seedlings cultivation practice, some studies showed that the exogenous uniconazole (S3307) could effectively promote the healing of grafted tomato seedlings (Meng et al., 2022). The exogenous IAA smearing at the graft junction could promote the transport capacity of xylem and phloem in grafted tomato seedlings (Cui et al., 2021). Some plant growth regulators, like an auxin-based plant growth regulator, naphthylacetic acid (NAA), commonly treat scions or rootstocks to accelerate graft healing. However, the physiological and molecular mechanism of graft healing by exogenous NAA application is not comprehensive. The main goals of the present study aimed to characterize the anatomical development stages of graft healing formation and clarify the physiological and transcriptomic changes in the graft junction of oriental melon grafted onto squash under NAA application, and provide a theoretical and practical basis for further improving the efficiency of commercial grafted melon seedlings cultivation.

Material and Methods

Experimental materials

In the present study, we took the oriental melon cultivar (YinQuan No.1, Cucumis melo var. Makuwa Makino) to graft onto the squash cultivar (ShengZhen No.1, C. moschata) by splicing graft method when the scion’s first-true-leaf fully expanded and rootstock’s cotyledon development stage, using the one-cotyledon method (Davis et al., 2008). During the grafting process, the grafted seedlings of scions dipped in the NAA solution (40 mg L−1) were the treatments (NAA), and dipping in the distilled water were the controls (CK). Grafted seedlings were transplanted in the nutritional bowl (12 cm ×12 cm) and moved into the healing chamber for grafted seedlings cultivation at Shenyang Agriculture University. The management methods of grafted seedlings for CK and NAA were consistent (Liu et al., 2017).

Paraffin sectioning and microscopy

The samples of 0.3−0.5 cm stem above and below the graft junction were fixed, softened, dehydrated, infiltrated, and embedded in paraffin during graft union development (Ribeiro et al., 2015). Transverse serial sections (≈10 µm thickness) were stained with pH4.4 toluidine blue (O’Brien, Feder & McCully, 1964) and mounted using synthetic resin (Permount). Sections were examined using a light microscope (Leica RM 2245, Wetzlar, Germany). After observing the paraffin section, the IL, CA, and VB stage of CK and NAA was respectively screened, and the corresponding samples carried out the other experiments for measuring the activities of enzymes involved in ROS scavenging (SOD, POD, PAL, and PPO), endogenous hormones contents (IAA, GA3, and CTK), and RNA-seq assay.

Determination of SOD, POD, PAL, and PPO activities

The graft junction tissues (≈0.5 g) of CK and NAA at the IL, CA, and VB stage with three biological replicates were ground with a pestle in an ice-cold mortar with 4 ml 50 mM phosphate buffer (pH 7.0). The homogenates were centrifuged at 12,000 rpm for 20 min at 4 °C, and the supernatant was used to measure enzyme activities with three technical replicates (Mishra et al., 2006). SOD activity was assayed by measuring its ability to inhibit the photochemical reduction of nitro blue tetrazolium. POD activity was measured as the increase in absorbance at 470 nm caused by guaiacol oxidation (Polle, Otter & Seifert, 1994). PAL activity was measured (Sánchez-Rodríguez et al., 2011). The reaction mixture was 0.4 ml of 100 mM Tris–HCl buffer (pH 8.8), 0.2 ml of 40 mM phenylalanine, and 0.2 ml of enzyme extract. The reaction mixture was incubated for 30 min at 37 °C, and the reaction was ended by adding 25% trichloroacetic acid. The absorbance of the supernatant was measured at 280 nm. PPO activity was spectrophotometrically determined at 398 nm (Olmos et al., 1997). The reaction mixture contained 2.8 ml 0.1% catechol solution and 0.2 ml enzyme extract in a total volume of 3 ml.

Determination of hormones content by ELISA

We sampled the graft junction tissues of CK and NAA at the IL, CA, and VB stage with three biological and technical replicates. The content of IAA, GA3, and ZR was measured by the Enzyme-Linked Immunosorbent Assay (ELISA), and the detailed protocol for determining the hormone content was previously described (Yang et al., 2001). GC-MS and HPLC validated the accuracy of ELISA kits (China Agricultural University).

RNA extraction, library construction, and sequencing

Total RNA extraction, library construction, and RNA-Seq were performed by Biomarker Technology Co. (Beijing, China, https://www.biocloud.net/). RNA concentration was measured using NanoDrop 2000 (Thermo Fisher Scientific, Waltham, MA, USA), and the detailed protocol was previously described (Xu et al., 2021). The raw sequencing data had been uploaded to the NCBI Sequence Read Archive (SRA) with the accession number PRJNA655799 and PRJNA689873.

Differential expression analysis

We respectively performed Melon (DHL92) v3.6.1 Genome and Cucurbita moschata (Rifu) Genome (http://cucurbitgenomics.org) to analyze the raw sequencing data of graft junction of CK and NAA at IL, CA, and VB stage. Differential expression analysis of two conditions/groups was performed using the DEseq. DEseq provides statistical routines for determining differential expression in digital gene expression data using a model based on the negative binomial distribution. The resulting p-values were adjusted using Benjamini and Hochberg’s approach for controlling the false discovery rate. Genes with an adjusted p-value <0.01 found by DEseq were assigned as differentially expressed (Zhang et al., 2018).

Enrichment analysis of GO enrichment and KEGG pathway

Gene Ontology (GO) enrichment analysis of the differentially expressed genes (DEGs) was implemented by the GO seq R packages based on Wallenius non-central hyper-geometric distribution (Young et al., 2010), which can adjust for gene length bias in DEGs. KEGG is a database resource for understanding high-level functions and utilities of the biological system, such as the cell, the organism, and the ecosystem, from molecular-level information, especially large-scale molecular datasets generated by genome sequencing and other high-throughput experimental technologies (http://www.genome.jp/kegg/). We used KOBAS software to test the statistical enrichment of differential expression genes in KEGG pathways (Mao et al., 2005).

Quantitative real-time PCR (qRT-PCR)

Some differentially expressed genes were selected to validate the accuracy of RNA-Seq data using qRT-PCR preparation with three biological and technical replicates for each sample conducted as described above. According to the manufacturer’s instructions, the first-strand cDNA synthesis kit was performed using a Prime-Script™ II First Strand cDNA synthesis kit (Takara Bio, Dalian, China). The primer sets (Table S1) for each unigenes were designed by Primer Premier 5.0. qRT-PCR was carried out on Yena Real-Time PCR System (qTOWER3/qTOWER3 touch, Analytik Jena, Jena, Germany) with SYBR Premix Ex Taq™ II kit (Takara).

Statistical analysis

The data were displayed with the mean ±standard error in triplicate and analyzed by a-way variance (ANOVA, SPSS 22.0 software). Significant analysis was performed by Duncan’s multiple range tests (p < 0.05). The figures were produced by PRISM 8.0 software.

Results

Effects of exogenous NAA application on histological changes of graft junction

A significant adhesion was observed in the graft junction of oriental melon grafted onto squash at 2 DAG (Figs. 1A, 1D). The isolation layer (IL)was generally formed by the protoplasm of destructed parenchyma cells and dead cells on the wound interface (Yang et al., 2016). The results showed that exogenous NAA application could not promote the formation of an isolated layer. With the graft junction development, the isolation layer gradually disappeared. Callus tissue (CA) provides a pathway for the communication between scion and stock (Wang & Kollmann, 1996). The graft junction with exogenous NAA treatment formed a callus tissue at 5 DAG (Fig. 1E), while the callus formation of CK was observed at 6 DAG (Fig. 1B). The vascular connection between the grafted partners was a mark of grafting success (Pina & Errea, 2005; Olmstead et al., 2006). At 8 DAG (Fig. 1F), the graft junction with exogenous NAA treatment formed vascular bundles (VB), and the new vascular bundle formation of CK occurred at 9 DAG (Fig. 1C). The results suggested that exogenous NAA application could shorten the graft healing process of oriental melon seedlings grafted onto squash.

Figure 1 Histological changes of graft junction using paraffin sectioning and microscopy method during graft union development.

(A and D), isolation layer stage (IL stage); (B and E), callus tissue stage (CA stage); (C and F), vascular bundles stage (VB stage). SC, scion; RT, rootstock; DAG, days after grafting.

Effects of exogenous NAA application on the activities of the related enzyme involved in ROS scavenging of graft junction during graft union development

In higher plants, mechanical wounding generates ROS production, and the antioxidant enzyme activities in the graft junction interface may be responsible for the degradation of the grafting zone (Aloni et al., 2008; Ślesak et al., 2008; Irisarri et al., 2015; Xu et al., 2015). Along with the graft union development, SOD activities of CK and NAA treatment first rose and then fell and presented the highest activity at the CA stage. SOD activity of NAA treatment was significantly higher than CK at the IL, CA, and VBstage, respectively (Fig. 2A). POD activity was significantly higher than CK at the CA stage and reached the highest activity under NAA treatment (Fig. 2B). In CK, POD activity was the highest value, and no significant difference compared with NAA treatment at the VB stage. As shown in Fig. 2C, NAA treatment increased PAL activity during the graft healing process. PAL activity was significantly higher than CK except for the CA stage and reached the highest activity at the VB stage. PPO activity had a similar trend in CK and NAA, which increased. Under NAA treatment, the value was significantly higher than CK at the IL stage. There were no significant differences in PPO activity at the CA and VB stages between CK and NAA treatment.

Figure 2 Effects of exogenous NAA application on SOD, POD, PAL, and PPO activities during graft union development.

(A) SOD; (B) POD; (C) PAL; (D) PPO. Different letters indicate significant differences (p < 0.05). Values are means ± SD, n = 3.

Effects of exogenous NAA application on endogenous hormones contents of graft junction during graft union development

Some major endogenous hormones, IAA, CTK, and GA, relate to callus formation development and vascular bundle reconnection (Nieminen et al., 2008; Mauriat & Moritz, 2009; Bishopp et al., 2011). Our results indicated that exogenous NAA application accelerated the graft healing process by promoting the content of three endogenous hormones at three critical stages. During the early stage of graft healing, a block in auxin basipetal transport was generated due to vascular damage, and the content showed a decreased trend. As the healing process was completed, auxin was accumulated in the wound interface and the content gradually increased. Under NAA treatment, the auxin content was lower than CK at the IL stage and was significantly higher than CK at the CA and VB stages (Fig. 3A). In the process of graft healing, the content of GA3 increased gradually.. When exogenous NAA was applied, the changing trend of GA3 content was consistent with CK and significantly higher than CK at the IL and VB stage (Fig. 3B), which was 1.25 times that of CK. The content of ZR gradually increased during the graft healing process and reached the highest value in CK at the VB stage. Under NAA treatment, the trend of ZR content was consistent with CK, and the ZR content was significantly higher than CK at the CA and VB stages, which was 1.3 and 1.2 times that of CK, respectively (Fig. 3C).

Figure 3 Effects of exogenous NAA application on endogenous hormone contents of IAA, GA, and ZR during graft union development.

(A) IAA; (B) GA3; (C) ZR. Different letters indicate significant differences (p < 0.05). Values are means ± SD, n = 3.

Identification of differentially expressed genes (DEGs) of graft junction under exogenous NAA application during graft union development

In order to further study the molecular mechanism of exogenous NAA application to regulate the graft healing process of oriental melon grafted onto squash, we performed transcriptomic analysis of graft junction tissue at the IL, CA, and VB stage under exogenous NAA application. We identified the DEGs in the melon genome (Fig. 4A) and Cucurbita moschata (Rifu) genome (Fig. 4B), respectively, according to the criteria of at least two-fold change and FDR<0.01. The 5324 DEGs were discovered by analyzing CK vs. NAA in the melon genome (Fig. 4A), with 1621 up-regulated and 1414 down-regulated at the IL stage; 261 up-regulated and 296 down-regulated at the CA stage; with 815 up-regulated and 917 down-regulated at the VB stage. 6602 DEGs were identified by analyzing CK vs. NAA in the cucurbita moschata (Rifu) genome (Fig. 4B), with 696 up-regulated and 914 down-regulated at the IL stage; with 512 up-regulated and 1,477 down-regulated at the CA; with 1,574 up-regulated and 1,429 down-regulated at the VB stage.

Figure 4 The numbers of DEGs of graft junction at the IL, CA, and VB stage under exogenous NAA application.

(A), CK vs. NAA in Melon genome; (B), CK vs. NAA in Cucurbita moschata (Rifu) genome.

Gene ontology and pathway enrichment analyses DEGs induced by exogenous NAA during graft union development

We used the Gene Classification System (GO) to further analyze the identified DEGs under exogenous NAA treatment at the IL, CA, and VB stage. The results showed that most DEGs were classified into three categories: biological process, cellular component, and molecular function (Fig. 5). At the IL, CA, and VB stage (Figs. 5A, 5C, 5E), there most highly enriched terms in the Melon genome were metabolic process (1,270, 206, 693 genes), single-organism process (1,142, 196, 609 genes) and cellular process (829, 153, 454 genes) within the biological process category; cell (804, 152, 509 genes), cell part (824, 124, 432 genes) and membrane part (801, 119, 424 genes) within the cellular component; binding (1,031, 186, 586 genes), catalytic activity (1,041, 200, 605 genes), and transporter activity (162, 42, 108 genes) within the molecular function category. Furthermore, at the IL, CA, and VB stage (Figs. 5B, 5D, 5F), there most abundant terms in the Cucurbita moschata (Rifu) genome were metabolic process (540, 836, 1,163 genes), single-organism process (453, 709, 847 genes) and cellular process (504, 768, 1,129 genes) within the biological process category; cell (321, 538, 843 genes), cell part (321, 538, 843 genes) and organelle (222, 366, 638 genes) within the cellular component; binding (395, 561, 769 genes), catalytic activity (494, 725, 845 genes), and transporter activity (79, 80, 114 genes) within the molecular function category.

Figure 5 GO functional classification and enrichment analysis of DEGs during graft union development.

(A, C, E) CK vs. NAA at the IL, CA, and VB stage using Melon genome. (B, D, F) CK vs. NAA at the IL, CA, and VB stage using Cucurbita moschata (Rifu) genome.

The DEGs were also subjected to KEGG pathway enrichment analysis. The top 20 pathways, which highest enrichment level based on the numbers and enrichment levels of the annotated DEGs, were shown in Fig. 6. The results were consistent with the results of GO functional analysis. It was noteworthy that plant hormone signal-transduction, phenylpropanoid biosynthesis, and phenylalanine metabolism were the overlapping pathways identified at the IL, CA, and VB stage under exogenous NAA treatment, respectively. At the IL stage (Figs. 6A, 6B), 53 DEGs, 18 DEGs (using Melon genome), and 33 DEGs, 20 DEGs (using Cucurbita moschata genome) were involved in plant hormone signal transduction and phenylpropanoid biosynthesis, respectively. At the CA stage (Fig. 6C) and VB stage (Fig. 6E), we found that 12 DEGs, 10 DEGs, and 34 DEGs, 21 DEGs were enriched in plant hormone signal transduction and phenylpropanoid biosynthesis using Melon genome. However, 45 DEGs, 20 DEGs (Fig. 6D), 38 DEGs, and 20 DEGs (Fig. 6F) were involved in phenylpropanoid biosynthesis and phenylalanine metabolism using Cucurbita moschata genome.

Figure 6 KEGG pathway enrichment analysis of DEGs during graft union development.

(A, C, E) CK vs. NAA at the IL, CA, and VB stage using Melon genome. (B, D, F) CK vs. NAA at the IL, CA, and VB stage using Cucurbita moschata (Rifu) genome.

Exogenous NAA activated hormone signal-transduction pathway during graft union development

Plant hormones of scion-rootstock communication are critical for successful graft (Melnyk et al., 2015). In our study, IAA, GA3, and ZR contents of graft junction increased under exogenous NAA application during graft union development (Fig. 3), and most DEGs were enriched in the hormone signal-transduction pathway (Fig. 6). So we analyzed the unigenes that participated in the hormone signal-transduction under exogenous NAA application (Fig. 7). At the IL stage, four unigenes encoding auxin efflux carrier (MELO3C019102.2, CmoCh11G003520, CmoCh04G021920, CmoCh11G003180), three unigenes encoding auxin response factors (ARFs) (MELO3C003768.2, MELO3C025777.2, MELO3C019801.2), five unigenes encoding AUX/IAA (MELO3C007691.2, MELO3C024699.2, MELO3C004382.2, CmoCh10G006330, CmoCh09G004620), two unigenes encoding GH3 (MELO3C008672.2, MELO3C017825.2) were significantly up-regulated in auxin signaling. And three unigenes encoding type-B ARR protein (MELO3C012031.2, CmoCh15G008650, CmoCh05G000700), one unigene encoding CRE1 (CmoCh15G009250) and two unigenes encoding AHP (MELO3C024439.2, CmoCh14G013260) in cytokinin signaling were also greatly up-regulated. At the CA stage, one unigene encoding ARF (MELO3C033303.2) in auxin signaling, one unigene encoding type-B ARR protein (CmoCh14G016360) and four unigenes encoding AHP (CmoCh14G016980, CmoCh17G002800, CmoCh14G013260, CmoCh06G014530) were significantly up-regulated. At the VB stage, one unigene encoding auxin efflux carrier (CmoCh07G011410), one unigene encoding ARF (CmoCh08G001220), three unigenes encoding AUX/IAA (MELO3C025308.2, CmoCh07G006010, CmoCh14G018090), four unigenes encoding GH3 (MELO3C016616.2, MELO3C007597.2, CmoCh03G009710, CmoCh16G003520) in auxin signaling, and one unigene encoding type-B ARR protein (CmoCh15G008650), two unigenes encoding AHP (MELO3C024439.2, MELO3C015359.2) were significantly up-regulated, respectively. However, we found that exogenous NAA application did not significantly activated unigenes expression involved in GA signaling.

Figure 7 Expression patterns of DEGs involved in hormone signal-transduction.

The values of log2 fold change were shown in the heat map.

Exogenous NAA application activated expression of genes involved in Reactive Oxygen Species (ROS) Scavenging during graft union development

The efficient antioxidant defense system plays a vital role in the graft healing process and is necessary for successful grafting. Under exogenous NAA application, we found many DEGs involved in ROS scavenging, including unigenes encoding peroxidase (POD), ascorbate peroxidase (APX), peroxiredoxin (Prx), superoxide dismutase (SOD), and phenylalanine ammonia-lyase (PAL) were activated during graft union development (Fig. 8). At the IL stage, eleven genes expression of POD, one gene expression of APX, two genes expression of Prx, one gene expression of SOD, and two genes expression of PAL were up-regulated by exogenous NAA application. At the CA stage, three genes expression of POD were up-regulated. Moreover, at the VB stage, five genes expression of POD, one gene expression of APX, two genes expression of Prx, two genes expression of SOD, and one gene expression of PAL were up-regulated.

Figure 8 Expression profiles of DEGs involved in ROS scavenging.

The values of log2 fold change were shown in the heat map.

Exogenous NAA application activated expression of genes related to vascular bundle formation during graft union development

It is well-known that the vascular bundle reconnection is a mark of successful grafting (Pina & Errea, 2005). In our study, exogenous NAA application accelerated the healing of oriental melon scion grafted onto the squash rootstock. In order to clarify the genes related to the vascular bundle formation process, including the cell elongation, vascular cell differentiation, and developing vascular cell trigger programmed cell death (PCD), we performed the heat map analysis of different unigenes expression (Fig. 9), such as expansion, tubulin, cellulose synthase, cinnamoyl-CoA reductase (CCR), hydrolytic enzyme (aspartic proteinase, cysteine proteinase), nuclease (exonuclease, ribonuclease), metacaspases and MYB transcription factors, using transcriptome data. Enzymes such as expansions are necessary for cell growth and cell wall architecture (Ye & Zhong, 2015). The results showed that six unigenes encoding expansion (MELO3C017181.2, MELO3C025095.2, CmoCh11G001730, CmoCh01G005150, CmoCh03G004350, CmoCh10G002140) were up-regulated at the VB stage. Tubulin is involved in cell expansion by guiding nascent cellulose microfibrils deposition during vascular development (Mendu & Silflow, 1993; Mo et al., 2018). We found that the expression of five tubulin genes (CmoCh03G009120, CmoCh07G010840, CmoCh14G000910, CmoCh07G006190,CmoCh16G003240) were also elevated by exogenous NAA application at the VB stage, but CmoCh03G009120, CmoCh07G010840, CmoCh14G000910, and CmoCh07G006190, except CmoCh16G003240, were down-regulated at the CA stage. We identified four unigenes, encoding cellulose synthase (MELO3C016270.2, CmoCh16G005790, CmoCh09G010200, CmoCh14G009470) implicated in cellulose synthesis, were significantly up-regulated at the VB stage. One unigene encoding CCR (MELO3C017061.2) involved in the phenylpropanoid pathway was up-regulated at the IL and VB stages. Furthermore, five unigenes encoding aspartic proteinase (MELO3C020328.2, CmoCh15G014930, CmoCh07G004000, CmoCh04G013980, CmoCh18G008610), two unigenes encoding cysteine proteinase (MELO3C027001.2, CmoCh03G010910), one unigene encoding exonuclease (CmoCh13G010360), and two unigenes encoding ribonuclease (MELO3C023673.2, CmoCh03G010620), all of them involved in xylogenesis (Iliev & Savidge, 1999; Courtois-Moreau et al., 2009; Iakimova & Woltering, 2017; Mo et al., 2018), were up-regulated at the VB stage. Meanwhile, one unigene encoding metacaspases (CmoCh11G014810) that participated in PCD was up-regulated at the VB stage. Additionally, sixteen unigenes encoding MYB transcription factors involved in the phenylpropanoid biosynthesis pathway were identified and significantly up-regulated by exogenous at the VB stage.

Figure 9 Expression profiles of DEGs involved in vascular bundle formation.

The values of log2 fold change were shown in the heat map.

Validation of RNA-Seq data by qRT-PCR

To verify the DEGs related to hormone signal and phenylpropanoid biosynthesis identified using RNA-Seq, we performed qRT-PCR assays with independent samples collected from graft junction tissues at different graft healing stages. The expression levels of these selected ten genes were identified between the two data sets (Fig. 10). The result showed that the expression of ten genes detected by qRT-PCR matched the trend of their FPKM value change in RNA-Seq (Table S2).

Figure 10 Verification of differentially expressed genes by qRT-PCR.

Different letters indicate significant differences (p < 0.05). Values are means ± SD, n = 3.

Discussion

Some reports showed that exogenous auxin application could promote callus formation (Sachs, 1981); Enrico et al., 2006). We found that exogenous NAA application accelerated the callus formation time and effectively promoted the time of vascular bundles connection of oriental melon scion grafted onto squash rootstock. Our results are consistent with the findings of Lu & Song (1999), who indicated that exogenous plant hormone application could accelerate the rate of plant graft healing.

During the graft union development, the antioxidant defense system, including SOD, PAL, POD, and PPO, was essential for the graft healing process (Gulen et al., 2002; Baxter, Mittler & Suzuki, 2014). PAL and POD enzyme was associated with lignin biosynthesis and tubular molecular formation (Duke, Hoagland & Elmore, 1980; Nakashima et al., 1997; Venema et al., 2008). In addition, PPO not only protected enzymes in plants but also played an important role in lignin biosynthesis. The higher the PPO activity, the longer the existence time of the graft interface between rootstock and scion, which affected the survival of graft healing (Lavee, 1989; Ali et al., 2006; Zhao et al., 2013). In our study (Fig. 2), the NAA treatment significantly enhanced SOD activities of graft junction during the graft union development (Fig. 2A). POD and PPO activities were remarkably increased by NAA treatment at the CA and IL stages, respectively. (Figs. 2B, 2D). PAL activities were significantly higher than the CK at the IL and VB stage under exogenous NAA application (Fig. 2C).

Furthermore, The induction of PPO activities increased some black and brown substances that form an isolated layer produced by the oxidation of phenolics when the plant was cut. The higher the PPO activities, the longer the graft interface’s existence time between rootstock and scion, which affected the survival of graft healing (Lopez-Gomez et al., 2007; Miao et al., 2019). In our study, NAA treatment had higher PPO activities compared with the control at the IL stage while had lower PPO activities at the CA and VB stage, potentially because the isolated layer disappeared, better formation of callus and the vascular bundle was beneficial to improve the healing speed and survival rate. As is well-known, ROS, the second messenger, regulates many biological processes, such as programmed cell death, cell cycle, and biotic or abiotic stress (Elhiti & Stasolla, 2015). And the higher expression of genes involved in ROS played an vital to the healing of grafted tomatoes (Xie, Dong & Shang, 2019). Our results indicated that the exogenous NAA treatment effectively removed the damage caused by ROS to the graft interface. Therefore, in order to regulate the graft healing process, it is very important to deeply explore and verify the genes related to ROS in the melon grafted onto squash.

IAA, CTK, and GAs are the primary hormones related to callus formation, vascular bundle development, and reconnection (Nieminen et al., 2008; Mauriat & Moritz, 2009; Bishopp et al., 2011). The appropriate concentration of IAA was added to the undifferentiated tissues, which could promote the formation of vascular tissues (Wetmore & Rier, 1963; Sachs, 1981; Aloni, Baum & Peterson, 1990). In our study, the IAA content decreased first and then increased during the graft healing process, consistent with Mo et al. (2018). When exogenous NAA was applied at the grafting interface, the content of IAA was significantly higher than CK at the CA and VB stage (Fig. 3A). Our results indicated that the increase of IAA content could promote the development of the xylem and phloem. ZR participates in cell division and vascular differentiation (Hejátko et al., 2009; Bishopp et al., 2011). Our results showed that the ZR content increased first and then decreased with the increase of grafting days in CK plants and reached the highest value at the CA stage. Under exogenous NAA treatment, the ZR content was significantly higher than CK at the CA stage, which promoted callus formation and accelerated the graft healing process (Fig. 3C). GAs can accumulate in developing xylem tissues of poplar trees (Israelsson, Sundberg & Moritz, 2005; Immanen et al., 2016). This finding coincided with our results of significantly increased GA3 content at the VB stage (Fig. 3B). When NAA was exogenous applied, the content of GA3 was significantly higher than CK at the VB stage. The possible reason is that the higher the GA3 content, the shorter the time for the vascular bundle formation. Our results indicated that after exogenous NAA application, IAA, ZR, and GA3 were higher than CK at the CA and VB stage, suggesting that endogenous regulation was performed under exogenous NAA application. Changes in hormone content affected the graft healing process of oriental melon scion grafted on squash rootstock.

RNA-seq technology has been used to explore the potential transcriptional regulatory mechanisms of the graft healing process (Cheong et al., 2002; Cookson et al., 2013; Liu et al., 2015; Chen et al., 2017; Melnyk et al., 2018; Mo et al., 2018; Xu et al., 2021). In order to further analyze the potential molecular networks of exogenous NAA regulating graft healing of oriental melon scion grafted onto squash rootstock, we performed the transcriptome analysis of graft junction at the IL, CA, and VB stage by CK vs. NAA using Melon genome and Cucurbita moschata (Rifu) genome, respectively, and deeply excavated the DEGs involved in hormone signal-transduction, ROS scavenging, and vascular bundle formation during graft union development (Figs. 7–9). Some reports showed that plant hormone signal-transduction, phenylpropanoid biosynthesis, and phenylalanine metabolism was crucial for graft healing (Qiu et al., 2016; Xu et al., 2017; Mo et al., 2018). Our study similarly found that plant hormone signal-transduction, phenylpropanoid biosynthesis, and phenylalanine metabolism were the overlapping pathways identified at the IL, CA, and VB stage by KEGG pathway enrichment analysis under exogenous NAA treatment.

Previous studies have shown that some genes involved in auxin, cytokinin, and gibberellin signaling are critical regulators for graft union development (Pils & Heyl, 2009; Cookson et al., 2013; Qiu et al., 2016; Mo et al., 2018). Auxin signaling was transmitted via transcriptional regulation of auxin early responsive gene families, including AUX/IAA, Gretchen Hagen 3 (GH3) (Shen et al., 2014; Feng et al., 2015). Several auxin-responsive genes were thought to be regulated during the graft healing formation at the transcriptional level (Zheng et al., 2010; Qiu et al., 2016). Auxin signaling via auxin response factors (ARFs) might be essential for graft union development (Hardtke & Berleth, 1998). Additionally, cytokinin played an essential role in cell division and vasculature differentiation via the two-component regulatory pathway to active type-B ARR transcription factors (Hardtke & Berleth, 1998; Nieminen et al., 2008; Hejátko et al., 2009; Pils & Heyl, 2009; Aloni et al., 2006; Ikeuchi, Sugimoto & Iwase, 2013). In our study, exogenous NAA application significantly promoted the expression of genes involved in auxin signaling, including genes encoding auxin efflux carrier (one melon gene, four cucurbita genes), encoding ARFs (four melon genes, one cucurbita gene), encoding AUX/IAA (four melon genes, four cucurbita genes), and GH3 (four melon genes, two cucurbita genes) during the graft healing process. Meanwhile, genes involved in cytokinin signaling, type-B ARR (one melon gene, three cucurbita genes), CRE1 (one cucurbita gene), AHP (two melon genes, four cucurbita genes), were up-regulated by exogenous NAA. However, we found no significant effects on gibberellin signaling by exogenous NAA application. Moreover, exogenous NAA had different impacts on the melon scion and squash rootstock during graft union development, although exogenous NAA could accelerate graft healing. Besides the hormones, an efficient antioxidant defense system was also crucial in achieving successful grafting (Aloni et al., 2008; Xu et al., 2015; Mo et al., 2018). In the present investigation, exogenous NAA not only promoted the expression of sixteen genes (ten melon genes, six cucurbita genes) encoding POD and two genes (one melon gene, one cucurbita gene) encoding APX but also accelerated the expression of three genes (three melon genes, one cucurbita gene) encoding Prx, two genes (one melon gene, one cucurbita gene) encoding SOD, and three genes (three melon genes) encoding PAL. Exogenous NAA application improved the abilities of ROS scavenging during graft union development.

Correspondingly, the new vascular tissue formation between scion and rootstock marks the success of grafting (Pina & Errea, 2005). This process included a complex physiological and biochemical reaction, such as cell elongation, vascular cell differentiation, and the development of vascular cells that trigger programmed cell death (Ye & Zhong, 2015). Under exogenous NAA application, the vascular bundle connection between melon scion and squash rootstock was earlier than the CK. Moreover, through the transcriptome analysis of CK vs. NAA at the IL, CA, and VB stage, most genes involved in vascular bundle formation were activated by exogenous NAA (Fig. 9). During cell elongation, expansion participated in cell growth and cell wall architecture (De et al., 2015). As expected, six genes expression (two melon genes, four cucurbita genes) were significantly up-regulated by exogenous NAA at the VB stage. Previous studies indicated that tubulin genes were involved in cell expansion and played an essential role in cell division and elongation (Mendu & Silflow, 1993; Mo et al., 2018). The expression of five cucurbita tubulin genes was enhanced by exogenous NAA at the VB stage. Differentiating vascular cells will conduct with the deposition of cellulose hemicellulose and lignin in the secondary cell wall after the cell elongation. We identified that four cellulose synthase genes (one melon gene, three cucurbita genes) are involved in cellulose synthesis (Suzuki et al., 2006), and one melon cinnamoyl-CoA reductase (CCR) gene is implicated in the phenylpropanoid biosynthesis pathway (Zhao et al., 2013) were up-regulated by exogenous NAA at the VB stage. We also identified sixteen MYB transcription factors (eight melon genes, eight cucurbita genes), and participated in the phenylpropanoid biosynthesis pathway as transcriptional regulators (Stracke, Werber & Weisshaar, 2001), and were significantly improved by exogenous NAA at the VB stage. Hydrolytic enzymes, including aspartic proteinase, cysteine proteinase, and nucleases (exonuclease, ribonuclease), were reported to operate the xylogenesis (Iliev & Savidge, 1999; Courtois-Moreau et al., 2009; Iakimova & Woltering, 2017). In our study, ten genes encoding them were detected with up-regulation under exogenous NAA application at the VB stage. We also detected one cucurbita gene encoding metacaspase involved in plant programmed death and vascular bundle differentiation (Sima et al., 2015; Mo et al., 2018) with higher expression under exogenous NAA application. Obviously, exogenous NAA application strongly triggered the expression of genes involved in vascular bundle formation during the graft healing process of oriental melon grafted onto squash.

Supplemental Information

Supplemental Information 1 The specific primers used in fluorescence quantitative PCR detection

Click here for additional data file.

Supplemental Information 2 FPKM value change of different genes in RNA-Seq

Click here for additional data file.

Supplemental Information 3 GO class of CK-IL vs NAA-IL in melon genome.

Click here for additional data file.

Supplemental Information 4 GO class of CK-IL vs NAA-IL in cucurbita genome.

Click here for additional data file.

Supplemental Information 5 GO class of CK-CA vs NAA-CA in melon genome.

Click here for additional data file.

Supplemental Information 6 GO class of CK-CA vs NAA-CA in cucurbita genome.

Click here for additional data file.

Supplemental Information 7 GO class of CK-VB vs NAA-VB in melon genome.

Click here for additional data file.

Supplemental Information 8 GO class of CK-VB vs NAA-VB in cucurbita genome.

Click here for additional data file.

Supplemental Information 9 KEGG pathway enrich dotplot of CK-IL vs NAA-IL in melon genome

Click here for additional data file.

Supplemental Information 10 KEGG pathway enrich dotplot of CK-IL vs NAA-IL in cucurbita genome

Click here for additional data file.

Supplemental Information 11 KEGG pathway enrich dotplot of CK-CA vs NAA-CA in melon genome

Click here for additional data file.

Supplemental Information 12 KEGG pathway enrich dotplot of CK-CA vs NAA-CA in cucurbita genome

Click here for additional data file.

Supplemental Information 13 KEGG pathway enrich dotplot of CK-VB vs NAA-VB in melon genome

Click here for additional data file.

Supplemental Information 14 KEGG pathway enrich dotplot of CK-VB vs NAA-VB in cucurbita genome

Click here for additional data file.

Supplemental Information 15 Raw data for heat map of expression patterns of DEGs involved in hormone signal-transduction

Click here for additional data file.

Supplemental Information 16 Raw data for heat map of expression profiles of DEGs involved in ROS scavenging

Click here for additional data file.

Supplemental Information 17 Raw data for heat map of expression profiles of DEGs involved in vascular bundle formation

Click here for additional data file.

Supplemental Information 18 Raw data of differential gene verification

Click here for additional data file.

Supplemental Information 19 Raw data of endogenous hormones contents

Click here for additional data file.

Supplemental Information 20 Raw data of the activities of the related enzyme involved in ROS scavenging

Click here for additional data file.

Abbreviations

NAA naphthylacetic acid

DAG day after grafting

DEGs differentially expressed genes

ROS reactive oxygen species

SOD superoxide dismutase

POD peroxidase

PAL phenylalanine ammonia-lyase

PPO polyphenol oxidase

IL isolated layer

CA callus

VB vascular bundle

IAA indole-3-acetic acid

GA gibberellin

ZR zeatin riboside

ELISA Enzyme-Linked Immunosorbent Assay

HPLC high-performance liquid chromatography

GC-MS gas-chromatography-mass spectrometry

RNA-seq transcriptome sequencing

qRT-PCR quantitative reverse transcription-polymerase chain reaction

Additional Information and Declarations

Competing Interests

Author Contributions

Data Availability

The authors declare there are no competing interests.

Chuanqiang Xu conceived and designed the experiments, performed the experiments, analyzed the data, prepared figures and/or tables, authored or reviewed drafts of the article, and approved the final draft.

Fang Wu conceived and designed the experiments, performed the experiments, analyzed the data, authored or reviewed drafts of the article, and approved the final draft.

Jieying Guo conceived and designed the experiments, performed the experiments, analyzed the data, authored or reviewed drafts of the article, and approved the final draft.

Shuan Hou performed the experiments, analyzed the data, authored or reviewed drafts of the article, and approved the final draft.

Xiaofang Wu performed the experiments, analyzed the data, authored or reviewed drafts of the article, and approved the final draft.

Ying Xin performed the experiments, analyzed the data, authored or reviewed drafts of the article, and approved the final draft.

The following information was supplied regarding data availability:

The raw sequencing data is available at NCBI Sequence Read Archive (SRA): PRJNA655799 and PRJNA689873. The raw measurements are available in the Supplementary Files.

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
