# Peer review of "Transcriptomic analysis and physiological characteristics of exogenous naphthylacetic acid application to regulate the healing process of oriental melon grafted onto squash"

_PeerJ, doi:10.7717/peerj.13980_

## Round 0.1 · original submission · Major Revisions

Please address all the comments.

Reviewer 1 ·

Basic reporting

In the manuscript entitled “Transcriptomic analysis and physiological characteristics of exogenous naphthylacetic acid application to regulate the healing process of oriental melon grafted onto squash”, authors investigated the effects of exogenous NAA application on histological changes, enzyme activities involved in ROS scavenging, endogenous hormones contents and transcriptome profiling of graft junction during graft union development.
The authors claimed that the exogenous NAA application could accelerate the graft healing process of oriental melon scion grafted onto squash rootstock, increase the SOD, POD, PAL, and PPO activities during graft union development and enhance the contents of IAA, GA3, and ZR in specific stage. The genes related to plant hormone signal-transduction, phenylpropanoid biosynthesis, phenylalanine metabolism, ROS scavenging system, and vascular bundle formation were differently expressed after NAA treatment. The manuscript provided a theoretical and practical basis for further improving the efficiency of melon grafting and the points of the manuscript is a very interesting topic regarding to commercial grafted melon cultivation.

Experimental design

1. In line 70-73 of Introduction: In the grafted seedlings cultivation practice, studies showed that applying exogenous plant growth regulators could effectively promote the healing of grafted seedlings, shorten the healing period of grafting, and improve the quality of grafted seedlings. The types of growth regulators and applied plants should be detailly described. In addition, related references should be cited.
2. In Materials and methods, the numbers of technical replicates should be indicated in determination of enzyme activities, endogenous hormones contents and qRT-PCR experiments. In present manuscript, just three biological replicates were introduced.
3. For the effects of exogenous NAA application on histological changes of graft junction, paraffin sections in all selected period should be provided to better understand the effect of exogenous NAA on graft healing process.

Validity of the findings

1. In line 179-180: SOD activity of NAA treatment was significantly higher than CK at the IL and CA stage (Figure 2A). The conclusion is incomplete. As shown in Figure 2A, SOD activity was also significantly increased in VB stage after NAA treatment.
2. For the effect of NAA application on GA3 content, the description in Results (line 198-201) and Discussion (line 372-373) parts were inconsistent with the data of Figure 3B.
3. In Figure 10, the RNA-Seq data of selected ten genes should also be provided to validate the correlation between RNA-Seq and qRT-PCR results.
4. Supplemental files should also be cited in manuscript.

Additional comments

1. The full name of the abbreviations such as IL, CA, VB, SOD, POD, PAL and PPO should be indicated when first mentioned.
2. There were numerous minor typographical errors in manuscript. For example, there were spelling mistakes in line 61 “uniqueities” and line 316 “tow”. Figure 10 was misquoted in line 299. In line 429, the exact stages are not specified. Therefore, the manuscript should be thoroughly reviewed.

·

Basic reporting

The manuscript entitled “Transcriptomic analysis and physiological characteristics of exogenous naphthylacetic acid application to regulate the healing process of oriental melon grafted onto squash” is an interesting study about the effects of exogenous naphthylacetic acid application on histological changes, enzyme activities involved in ROS scavenging, endogenous hormones contents and transcriptome profiling of graft junction during graft union development.”The authors have reported that application of naphthylacetic acid could accelerate the graft healing process of oriental melon scion grafted onto squash rootstock, increase the SOD, POD, PAL, and PPO activities during graft union development and enhance the contents of IAA, GA3, and ZR in specific stage Moreover, the exogenous NAA application signiûcantly promoted the expression of genes involved in the hormone signal-transduction pathway, ROS scavenging system, and vascular bundle formation.”. The manuscript clearly describes the possibilities to improve the efficiency of melon grafting and the points of the manuscript is a very interesting topic regarding to commercial grafted melon cultivation.

Experimental design

Materials and methods: The paraffin sections in all selected period could help to understand the effects of NAA application.

Validity of the findings

1. Figure 2 could be better discussed and conclusions based on the resuls can be improved.
2. The RNA-Seq data of selected ten genes could better validate the correlation between RNA-Seq and qRT-PCR results.

Additional comments

1. The full name of the abbreviations should be used when mentioned for the first time in manuscript.
2. Information about supplementary material should be mentioned in main manuscript.
3. Typographical errors should be removed before submission of revision.

---

## Round 0.2 · accepted · Accept

Please review for potential grammatical corrections.

Reviewer 1 ·

Basic reporting

This manuscript has been thoroughly reviewed and corrections have been incorporated.

Experimental design

Sufficient.

Validity of the findings

Robust.